# Effects of a Resistance Training Protocol on Physical Performance, Body Composition, Bone Metabolism, and Systemic Homeostasis in Patients Diagnosed with Parkinson’s Disease: A Pilot Study

**DOI:** 10.3390/ijerph192013022

**Published:** 2022-10-11

**Authors:** Alessandra Amato, Sara Baldassano, Sonya Vasto, Giuseppe Schirò, Chiara Davì, Patrik Drid, Felipe Augusto Dos Santos Mendes, Rosalia Caldarella, Marco D’Amelio, Patrizia Proia

**Affiliations:** 1Sport and Exercise Sciences Research Unit, Department of Psychological, Pedagogical and Educational Sciences, University of Palermo, 90128 Palermo, Italy; 2Department of Biological Chemical and Pharmaceutical Sciences and Technologies (STEBICEF), University of Palermo, 90128 Palermo, Italy; 3Department of Biomedicine, Neurosciences and Advanced Diagnostics (Bi.N.D.), University of Palermo, 90127 Palermo, Italy; 4Faculty of Sport and Physical Education, University of Novi Sad, 21000 Novi Sad, Serbia; 5Graduate Program in Rehabilitation Sciences, University of Brasília, Brasília 72220-275, Brazil; 6Department of Laboratory Medicine, “P. Giaccone” University Hospital, 90127 Palermo, Italy

**Keywords:** physical activity, neurodegenerative disease, resistance training, bone resorption, osteogenesis, osteoporosis, Parkinson’s disease

## Abstract

Parkinson’s disease (PD) is a neurodegenerative disorder characterized by motor impairments and it is correlated with loss of bone mineral density. This study aimed to analyze the effects of resistance training on bone metabolism, systemic homeostasis, body composition, and physical performance in people with PD. Thirteen subjects (age 64.83 ± 5.70) with PD diagnosis were recruited. Participants performed neuromuscular tests, body composition assessment, and blood sample analysis at baseline, and after an 11 weeks-training period. Each training session lasted 90 min, three times a week. The participants had significant improvements in the timed up and go (*p* < 0.01), sit to stand (*p* < 0.01), dominant peg-board (*p* < 0.05), dominant foot-reaction time (*p* < 0.01), and functional reach tests (*p* < 0.05). They showed better pressure foot distributions in the left forefoot (*p* < 0.05) and hindfoot (*p* < 0.05) and increased cervical right lateral bending angle (*p* < 0.05). The protocol affects bone metabolism markers osteocalcin (*p* < 0.05), calcium (*p* < 0.01), PTH (*p* < 0.01), the C-terminal telopeptide (CTX) (*p* < 0.01), and vitamin D (*p* < 0.05). Eleven weeks of resistance training improved manual dexterity, static and dynamic balance, reaction time, cervical ROM, and reduced bone loss in people with PD.

## 1. Introduction

Bone metabolism is defined as the balance between resorption activity, by osteoclasts, and the deposition of new bone matrix by osteoblasts. It is regulated by the endocrine signaling of hormones that regulate calcium, including parathyroid, calcitonin, and 1α, 25-dihydroxy vitamin D3, as well as by mechanical stimuli [1]. The alterations of resorption-formation mechanism balance are associated with pathologies characterized by motor impairment [2] as in Parkinson’s disease (PD), in which motor symptoms are associated with a decrease in bone mineral density [3] (Hart, Meehan, Bae, d’Hemecourt, & Stracciolini).

Parkinson’s disease (PD) is a neurodegenerative disorder, it increases in prevalence with age, causes motor and non-motor symptoms such as depression/psychosis, but also emotional and communicative changes like the typical “facial masking” and emotional speech (i.e., dysarthria), that could result in “social symptoms” such as stigma, dehumanization, and loneliness, with a consequent worst quality of life [4].

To classify PD, neurologists use the visual examination of motor tasks and semi-quantitative rating scales divided into five stages by Hoehn–Yahr [5]. The incidence of PD cases in 2019 was 1.1 million camp [6]. The economic burden of Parkinson’s was estimated to be USD 52 billion in 2017 in the USA, but these costs are underestimated. This is because intervention to delay disease progression, and the incidence and alleviation of symptoms, may reduce the future economic cost for society, patients, caregivers, and payers [7]. To make the diagnosis, the subject must present a distal resting tremor of 3 to 6 Hz, rigidity, bradykinesia, and asymmetrical onset [8]. According to the AAN guideline in 2002 [9], therapeutic options to relieve motor symptoms in the early stages of PD are based on enhancing dopaminergic tone with levodopa, monoamine oxidase inhibitors, dopamine agonists (DAs), or a combination thereof. The choice of treatment is influenced by the potential neuropsychiatric adverse effects associated with DAs. In fact, the recent AAN Issues Guideline for Treatment of Early Parkinson’s Disease [10] recommends the treatment of motor symptoms with levodopa medications over treatment with dopamine agonists more than DAs.

Physical activity is a key stimulus for symptoms management, it also produces lactate and myokines that contribute to brain health [11,12,13]. Several factors influence bone metabolism during physical activity, such as nutrition, as meals influence bone resorption through mechanisms still under study, such as the secretion of gut hormones [14,15,16,17,18,19,20,21,22]. However, this mechanical stimulus must have a specific range within which the bone tissue response is activated, specifically from 50 to 100 microstrain (με), defined as minimum effective strain (MES) [23].

Exercise seems to be effective to stimulate bone formation if it has certain characteristics, such as impact with the ground or other surfaces [15]. In addition, a mechanism that explains the importance of mechanical loading is that this loading, through mRNA stimulation, regulates the activity of osteogenic and bone resorption factors while maintaining the properties of the tissue [24]. There are receptors such as those of collagen, and mechano-sensitive integrins present in connective tissues that are activated exclusively upon mechanical stimulus [25]. Therefore, the reduction or complete absence of mechanical stimulus causes alteration and loss of bone tissue, predisposing to diseases related to it. The loss of mechanical stimulus is the main cause of bone disease in PD due to decreased daily life physical activity. Typically, people with PD frequently adopt a sedentary lifestyle as a result of their reduced postural stability, gait disturbances, and decreased strength. However, reduced physical activity, although the most obvious, is not the only cause of reduced bone mineral density (BMD) in PD: calcium metabolism regulatory hormones and vitamin D are extremely deficient in PD patients [26,27]; high-dose treatment with the dopamine precursor, levodopa, is the current standard drug for PD, and is associated with low BMD due to the deleterious effects of homocysteine (Hcy), a toxic metabolite that inhibits the bone formation [1,26,28,29]. Moreover, it has been shown that one of the possible causes of the onset of PD is genetic [30,31,32]. Some genes whose altered expression is linked to PD have been identified: the “Parkinson-designated genes” (PARKS). The PARKS are expressed not only in the brain but also in other tissues including bone cells [33]. They play a key role in bone tissue homeostasis, they act on osteoblastic differentiation of bone marrow stromal cells (BMSCs), they have a role in osteogenesis and osteoclast genesis, and they indirectly contribute to increasing bone thickness [30,31,32,34,35,36,37,38].

Beyond drug therapy, few or no “tools” have been identified in the literature that can help to reduce the resulting BMD loss in people with PD, including which exercise would be best suited for this purpose. To date, there are no studies that analyze the characteristics that physical exercise could have to better stimulate bone metabolism in people with neurodegenerative disease. Recent literature tends to simply associate daily physical activity with BMD through the use of accelerometers or data collected by self-report questionnaires [39,40]. Consequently, the purpose of our study was to evaluate the effects of a resistance training protocol on physical performance, body composition, bone metabolism, and systemic homeostasis. Additionally, this study intends to pave the way to understand whether physical exercises could be supportive of drug therapy to slow and manage the symptoms of disease progression and improve the quality of life of people with Parkinson’s disease.

## 2. Materials and Methods

We recruited 13 subjects (age 64,83 ± 5,70) diagnosed with Parkinson’s disease according to the criteria of the “United Kingdom Parkinson Society Brain Bank” [41]. All patients were recruited from the neurological center “Paolo Giaccone” University of Palermo, by a call in October 2021, from patients being treated at that hospital facility. Of the 13 patients, 11 (*n* = 84.6%) were male and the median disease duration in years was 5.7 *±* 3.08. The inclusion criteria were: (1) Parkinson’s diagnosis; (2) no dementia; (3) more than 50 years old; (4) self-sufficient walking; (5) being able to carry out our motor performance tests independently; and (6) attending all the training sessions.

One subject dropped out because he did not attend all the training sessions. Twelve subjects completed all training sessions and assessments. The participants performed motor performance tests, postural and static analysis, body composition analyses, and blood sample analyses at baseline (T0) and after 11 weeks of training (T1). They joined and completed the short-term intervention study with excellent compliance. The “Waterloo Footedness questionnaire (WFQ-R)” was administered to determine their dominant leg [42]. All patients were evaluated with the Hoehn and Yahr (H&Y) [43] staging and the Unified Parkinson’s Disease Rating Scale (UPDRS) [44]. To exclude patients with dementia, those who scored less than 21 on Montreal cognitive assessment (MoCA) [45] test were excluded from the current study. The Levodopa Equivalent Daily Dose (LEDD) [46] was also calculated for each patient to take into account the daily therapeutic load.

### 2.1. Motor Performance Tests

All the motor performance tests were performed at the Sport and Exercise Sciences Research Unit laboratory at the University of Palermo (Italy). Each participant was asked to come early in the morning. After the motor evaluations, within 2-3 days, the subjects would go to the neurology center in Palermo to have their blood drawn. All initial evaluations were done in November 2021.

#### 2.1.1. Postural and Static Analysis

The postural (also known as stabilometry) and the static (also known as baropodometry) analyses were performed with the FreeMed posturography system (produced by Sensor Medica, Guidonia Montecelio, Roma, Italy).

For the postural analysis, the subjects stood on the platform under two conditions: open (OE) and closed eyes (CE) for 51.2 s each with an angle of 30° between the feet. We have taken into consideration the Sway area (cm^2^), the area which contains the trajectory of the center of pressure (COP), the sway paths length (mm), the length of the distance traveled by the COP, and the Sway average speed (mm/s), the average speed at which the COP moves in its whole trajectory with OE and CE. As regards the static analysis, all participants stood on the platform for 10 s with the most comfortable position of the feet and the following parameters were evaluated: the foot area (cm^2^), forefoot, and hindfoot load (%) for the left and right foot.

#### 2.1.2. Cervical ROM

The cervical range of movement (ROM) was evaluated with an accelerometer (Sensor Medica) in order to measure the lateral rotation (°), lateral inclination, flexion, and extension (°).

#### 2.1.3. Timed Up and Go

The Timed Up and Go (TUG) test is a simple test that enables the measurement of the level of functional mobility and dynamic balance skills. It measures the time it takes a person to get up from a chair, walk 3 m, turn around, go back to the chair, and sit down again. A score of 10 s or less indicates normal mobility; times between 11 and 20 s are within normal limits for elderly with frailty or disabled patients; times longer than 20 s indicate that the person needs external assistance. A score greater than 30 s suggests that the person has a high risk of fall [47]. Alternatively, a recommended reference value as the limit between a normal capacity and one below normal is 12 s [48].

#### 2.1.4. Five Times Sit to Stand (5TSTS)

For the standardized administration of the 5TSTS, we followed the Academy of Neurologic Physical Therapy guideline [49]. The participant was asked to get up and sit back in the chair five times. The time taken to complete the test was recorded.

#### 2.1.5. Functional Reach Test

This test evaluates the static balance through maximal forward reach from a fixed base of support [50]. For the starting position, the subject stood with the dominant upper limb side of the body against the wall and the dominant upper limb extended at a 90° angle between the torso and the limb. The difference between arm’s length and maximal forward reach is measured, the test was repeated three times and was considered the best measure.

#### 2.1.6. Grooved Pegboard Test

The grooved pegboard test evaluates manual dexterity, i.e., the skills that allow us to coordinate the hands and fingers to manipulate objects essentially in daily life [51]. This ability is often compromised in neurological diseases. We used the Lafayette Instrument Grooved Pegboard Test tool. At the start signal, the subject placed pegs from left to right for the right-hand task and in opposite direction for the left hand. The dominant hand trial is administered first, followed by the nondominant hand trial. The time was measured with a digital chronometer. All participants performed the test two times for both dominant and nondominant hands, but we took into consideration just the results of the dominant limb.

#### 2.1.7. Handgrip Test

The Handgrip test measures the maximum strength of the muscles of the hand and forearm through the use of a digital dynamometer [52]. The subject holds the handgrip keeping the arm extended at the bodyside; at the operator’s signal, the participant must compress the handgrip as tightly as possible, for 3 s. The test must be repeated three times for each limb. The dynamometer used has three calibrations expressed in kg: 20 (for children), 40 (for women), and 80 (for men). The best result of the three trials for each limb was considered.

#### 2.1.8. Eye-Hand Reaction and the Foot-Reaction Time Test

We performed the eye-hands reaction test and the foot-reaction time test to evaluate the reaction skills. The tests are based on the measurement of the time interval between the presentation of a visual or acoustic stimulus and the execution of a response (pressing a specific button or lifting the foot off the floor). The tests were performed in the “simple task” condition, in which there was only one stimulus to which must be given only one answer. For the eye-hand reaction test, the participant was seated in a chair with the dominant limb placed on the table in front (90° between arm and forearm) where the computer was placed, with the index finger resting on the answer button. The subject had to press the button with the dominant limb once the light on the screen turned green. The test includes five evaluations, the average was calculated from the five scores obtained, expressed in seconds. To perform the foot-reaction time test we used the optical system OPTOJUMP (MICROGATE) and the OPTOJUMP NEXT software. The subject was standing with only one limb inside the platform, the OPTOJUMP NEXT software emitted a sound stimulus after which the subject had to lift the foot completely off the floor as quickly as possible and put it back in the shortest possible time. The test was carried out alternately with the right and left limbs for three times. The average of the reaction times recorded by the software was calculated to obtain the final value for each subject.

### 2.2. Body Composition Analysis

We analyzed the body composition by electrical bioimpedance measurement at 50 kHz, single frequency (BIAlight, DS MEDICA) to evaluate the following parameters before and after the training period: body mass index (BMI), total body water (TBW) (l), free fat mass (FFM) (Kg), fat (Kg), skeletal muscle mass (SM) (Kg), skeletal muscle mass index (SMI) (kg/m^2^), and basal metabolic rate (BMR) (Kcal). The BIA was performed fasting, the subject was lying for 10 min before starting the assessment with the right hand and foot bare to allow the electrodes to be attached and without metal objects on him.

### 2.3. Blood Sample Analysis

A venous whole blood sample was collected in VACUTAINER tubes and EDTA-K3 between 7:00 and 8:00 a.m. [53,54]. The blood sample was collected fasting at T0 and T1, in order to monitor the changes in bone markers and systemic homeostasis. Whole blood samples were collected in tubes without any anticoagulant and fractionated by centrifugation at 1300× *g* for 15 min at room temperature to obtain serum. The serum was used to measure the following chemical markers using automated procedures according to standard commercially commercial assays from Roche Diagnostics performed on the Roche COBAS c501: glucose, insulin, total cholesterol, HDL-cholesterol, triglycerides, triiodothyronine (FT3), thyroxine (FT4), thyroid stimulating hormone (TSH). To evaluate insulin resistance and insulin sensitivity we used a specific computer model described in [14,55] updated methods from Wallace et al. (2004) and Ghasemi et al. (2015) [56,57]. Serum samples were analyzed for C-terminal telopeptide (CTX), a marker of bone resorption, osteocalcin (also referred to as bone-GLA-protein, BGP), a marker of bone formation, and the markers of bone metabolism parathyroid hormone (PTH), calcitonin, albumin-adjusted calcium (Aa calcium), and Vitamin D.

BGP, PTH, CTX, and Vitamin D, were measured by electro-chemiluminescence immunoassay (ECLIA) following the same method used in Vasto et al. [15], and fluctuations in protein concentrations were ‘corrected’ to give an albumin-adjusted calcium (Aa calcium) value using specific equation [15].

### 2.4. Training Protocol

The training period ran from December 2021 to February 2022. The training protocol lasted 11 weeks tri-weekly, it was performed at the participant’s home using digital platforms with the help of two physical exercise experts. Each training session lasted 90 min and had three stages: the warm-up, the main workout, and the cool-down. The warm-up and the cool-down were the same for the entire duration of the study. The warm-up included breathing exercises and mobility exercises for the scapula-humeral, ankle, and hip joints, and the cervical spine and pelvic region. Each movement was performed for 10 repetitions or (if the movement was unilateral) five repetitions on each side with a rest period between 30″ and 40″ for each set. The difficulty of the exercises was increased with the use of different tools from one week to the next (Table 1). The cooldown included stretching exercises for the cervical spine, torso, upper limbs, and lower limbs. Each stretch position was maintained for 15 s. After the warmup, the main workout included three different weekly sessions (A, B, and C) of resistance training with intensity progressively increasing. Figure 1 is a general representation of each exercise performed by participants. The protocol, with the progression and the tools used, is detailed in Table 1.

### 2.5. Ethics

The principles expressed in the Declaration of Helsinki were met in our study. The ethics committee commission from the University of Novi Sad Faculty of Sport and Physical Education, gave us the following approval number:46-06-02/2020-1 Novi Sad, Serbia. All participants were informed about the study before the start and they signed written informed consent.

## 3. Statistical Analysis

We used the Shapiro–Wilk test to assess the normality of data. To analyze the difference between the mean of the variables that resulted in not being normally distributed we performed the non-parametric test Wilcoxon rank and we used the paired *t*-test to analyze the differences between the normally distributed variables. Pearson’s coefficient was calculated to analyze the correlations between bone metabolism markers. An r value < 0.5 was considered with a low correlation, values between 0.5 and 0.7 moderate correlation, and values > 0.7 indicated a strong correlation. Statistical analysis was performed with the R-Studio (© 2009–2022 RStudio, PBC) and the SPSS (IBM SPSS Statistics, version 23, Armonk, NY, USA) software.

## 4. Results

The anthropometric and clinical characteristics of patients are summarized in the following table (Table 2).

### 4.1. Motor Performance Test and Body Composition Analysis

The changes in motor performance and body composition variables before and after the training time are shown in Table 3 as mean and standard deviation. We noted an improvement in the following tests: timed up and go (s)(V = 78, *p*-value = 0.00); sit to stand test (s) (V = 75, *p*-value = 0.00); dominant peg-board test (s) (V = 68, *p*-value = 0.02); dominant foot-reaction time (V = 76.5, *p*-value = 0.00). Functional reach test (t = −3.13, df = 11, *p*-value = 0.01) and a change in pressure feet distribution: left foot area (cm^2^) (V = 58, *p*-value = 0.03); left forefoot load (%) (t = 3.44, df = 11, *p*-value = 0.01); left hindfoot load (%) (t = −3.44, df = 11, *p*-value = 0.01). Additionally, the R lateral bending (°) changed significantly (t = −2.53, df = 11, *p*-value = 0.03).

### 4.2. Blood Sample Analysis

Changes between T0 and T1 in all parameters analyzed with blood analysis are shown in Table 4.

There were no changes after 11 weeks of resistance training in lipid and glucose metabolism. Additionally, FT4 and TSH were not affected, while there was an increase in FT3 values (*p* = 0.00) suggesting that it impacts thyroid function. As regards bone markers we had an increased value for the osteocalcin (*p* = 0.01), PTH (*p* = 0.00), CTX (*p* = 0.00), and calcium (*p* = 0.00) concentration and a reduction in vitamin D (*p* = 0.04) (Table 4).

Correlation analysis between T0 and T1 for bone metabolism markers shows a strong positive correlation (red circle Figure 2) between CTX T1 with BGP T0 (r = 0.66, *p* < 0.05) and with BGP T1 (r = 0.73, *p* < 0.01). Correlations are shown in Figure 2.

## 5. Discussion

In the literature, the effects of resistance training in subjects with Parkinson’s disease have been evaluated in many studies [58,59,60,61,62,63,64]. More specifically, some of these evaluate the effect of this training on respiratory capacity [65,66,67,68,69], on the risk of falls and balance camp [70,71,72,73,74,75,76,77], and on the quality of life and depression camp [78,79,80]. However, none of them evaluates the effect of resistance training on the bone metabolism of these subjects. Individuals with PD have a high predisposition to lose bone tissue, and the lack of scheduled exercise contributes to this [26,27,33]. Since resistance training results in an increase in site-specific bone density more than other training [81,82,83], the purpose of this study was to evaluate the effect of home-based strength training on physical performance, body composition, bone metabolism, and systemic homeostasis, in people with Parkinson’s disease. In the context of evaluating the effect of this training on physical performance, we also wanted to assess whether neck stiffness, a hallmark of PD [84], varies as a result of resistance training, because neck rigidity seems to be caused by the impossibility of voluntary control of the long-latency proprioceptive reflexes [85] and training is associated with improved corticospinal plasticity for motor learning [86,87,88,89]. Our results have shown, although in absolute value, an improved trend in all movements of the cervical ROM with a statistically significant improvement in the right lateral bending (°) (*p*-value = 0.03). The axial tone was related to functional performance. Particularly, Frazen et al. showed how the neck tone is strongly correlated with balance and mobility disorders in individuals with PD [90] and our resistance protocol influenced balance, evident in static analysis: it shows a better feet pressure distribution (%) between forefoot and hindfoot at T1, significant for the left foot which perfectly meets the reference parameters of weight and maximum pressure distribution [91] and the load distribution (%) for the right foot after the training period is closer to the reference parameters than T0 (forefoot: *p* = 0.01; hindfoot: *p* = 0.10) [91]; the physiological pressure distribution of body weight at the feet is 60% on the rearfoot and to 40% midfoot and forefoot [91]. Alterations of the connective tissue, in age atrophy or diseases such as Parkinson’s, which see an altered distribution of plantar pressure, modify the normal characteristics of the forefoot [91]. People with Parkinson’s disease have impaired control of their balance during gait and static posture, often related to injuries resulting from falls and consequently a worse quality of life. Tsakanikas et al. (2021) showed that the spatial static distribution of the center of pressure can provide important information about the status of the patient both for static balance and gait monitoring [92]. The changes on feet pressure distribution affected, with significant improvements, the functional tests TUG test (dynamic balance), (T0 = 12.47± 2.69, T1 = 9.29± 2.16, *p* = 0.00), and functional reach test (static balance), (T0 = 31.37± 7.06, T1 = 38.73 ± 5.34, *p* = 0.01). Therefore, we hypothesized that reprogramming of postural tone, after the strength training period, could influence the pressure distribution at the feet and thus balance, and this to influence the results of functional tests “TUG” and “functional reach”.

Several studies found that people with Parkinson’s disease needed more time to perform reaction time tests compared to healthy controls [93,94]. Furthermore, Morrison et al., with a similar reaction test for upper and lower limbs, found a correlation between the risk of falling and slowing reaction time in Parkinson’s disease [95]. However, this skill could be trained and improved by simple or dual-task practice as demonstrated by recent studies [96,97]. Nevertheless, no studies so far have shown the effect of strength training on reaction time and, consequently, by reducing the reaction time, its possible impact on the risk of falls. In our study, the reaction time assessment, both to visual and acoustic stimulus, had an improving trend in the eye-hand reaction test (s) (from T0 = 0.54 ± 0.36, to T1 = 0.41 ± 0.11, *p* = 0.07) and non-dominant foot-reaction time (s) test (T0 = 0.83 ± 0.28, T1 = 0.61 ± 0.09, *p* = 0.05). A significant decrease in reaction time to the acoustic stimulus was found in the dominant foot-reaction time (s) test (T0 = 0.72 ± 0.15, T1 = 0.61 ± 0.08, *p* = 0.00).

Manual dexterity is the skill that allows us to manipulate objects thanks to eye-hand coordination and it is a mark of motor function [98]. Usually, manual dexterity is evaluated as the time to complete the pegboard test (s) that in our study was reduced from 137 ± 49.61 (T0) to 126.87 ± 51.59 (T1) seconds.

It is established that people with PD have a worse pegboard test performance than healthy adults of a similar age group [99]. The increase in the time it takes to complete the pegboard test is correlated with a declining psychomotor processing speed indicator of pathophysiological changes in top-down visual and motor control pathways. The pegboard test is also a predictor of the activities of daily living dysfunction [100]. In addition, our results show that the subjects after the 11 weeks of training passed from a score of 21.86 ± 8.52 (s) at T0 to a 14.19 ± 4.22 (s) (*p* = 0.00) in the 5× sit-to-stand test. It is demonstrated that 16 s to complete 5TSTS discriminates the fallers from the non-fallers, particularly more than 16 s indicating risk for falls, as mentioned before after the training period subjects reduced their 5TSTS under 16 s [101].

However, this training for 11 weeks does not change body composition parameters (Table 3) and this could be useful for these subjects who usually tend to quickly reduce their weight and their lean mass with the progression of the disease [102,103,104,105]. Regarding bone metabolism, our results suggest that could be an influence of resistance training on bone turnover, positive for osteocalcin (formation marker), which from 21.08 ± 6.66 (μg/L) at baseline increases significantly (*p* < 0.05) in the post-training period to 23.58 ± 6.58 (μg/L). However, this increased formation activity, also evidenced by an increase in calcium (*p* < 0.01), seems to be balanced by the concomitant increase in reabsorption markers CTX (*p* < 0.01), PTH (*p* < 0.01), and the reduction at T1 of Vitamin D (*p* < 0.05). In contrast, the correlation (r = 0.727) at T1 between CTX T1 and osteocalcin T1 indicates that this process is in balance and corresponds to the normal balanced bone turnover process. Thus, physical activity has a positive effect because meanwhile the two processes of formation and resorption are matched as they should in physiological conditions and there is no increased tendency for resorption as there is in disease conditions such as PD. In addition, besides the increase in osteocalcin and calcium, indicators of bone formation, the increase in PTH after physical activity may indirectly be indicative of bone formation [16]. However, the role of PTH in response to physical activity is still unclear even more so in the presence of subjects with pathology.

## 6. Limitations

This study has limitations: first of all, the lack of a control group that did not perform physical activity, although we evaluated each of the patients at baseline and at the end of the training, in future studies a control group is necessary. Another limitation is the small number of participants. The training set and the pilot nature of the project, made us unable to work with a larger number of subjects. Therefore, studies with a larger number of subjects are required. Another limitation was the lack of follow-up. We evaluated the effect of resistance training on bone metabolism in PD after 11 weeks of training. Given the chronic nature of the disease we do not know, by stopping the activity, for how long the benefits of the intervention were present and if by continuing it longer they could further improve their condition. Once again further investigations are required.

## 7. Conclusions

In conclusion, this resistance training for people with Parkinson’s is effective in improving physical performance and this is evident in the improvements of functional tests and would result in better daily living functions.

Additionally, this protocol, with this frequency and progression, seems to affect bone metabolism, certainly not increasing resorption at the expense of formation. Indeed, these two processes are shown to be in balance. Therefore, home-based resistance training could be a strategy to support drug therapy to prevent and reduce BMD loss associated with PD, maintain healthy body composition, and improve functional physical performance.

## Figures and Tables

**Figure 1 ijerph-19-13022-f001:**
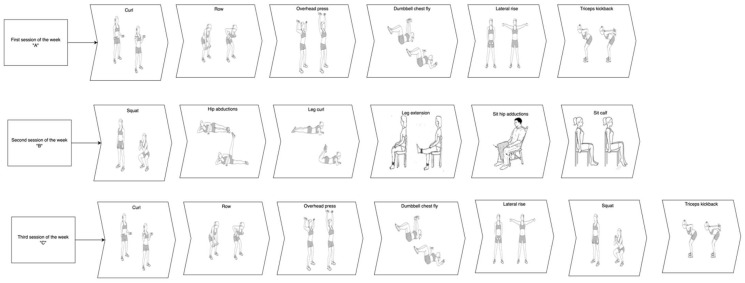
Graphical representation of each exercise performed by study participants in the three different weekly sessions. Note, each exercise could be performed with the support of a chair if the participant needed it.

**Figure 2 ijerph-19-13022-f002:**
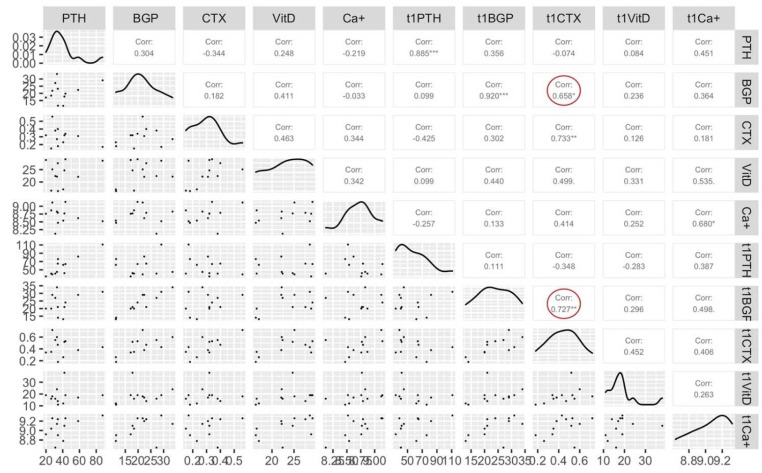
Correlations between all bone markers at baseline (T0) and after 11 weeks of training (T1). “*” = significative difference (*p* < 0.05), “**” = significative difference (*p* < 0.01) between T0 and T1 for Pearson’s correlation. PTH = parathyroid hormone, BGP = bone-GLA-protein (osteocalcin), CTX = C-terminal telopeptide, VitD = vitamin D, Ca+ = calcium.

**Table 1 ijerph-19-13022-t001:** Training sessions (A, B, and C) exercises and progressions.

Sessions	Weeks of Training				
A	1 Set × Reps ^tool^	2, 3 Set × Reps ^tool^	4–6 Set × Reps ^tool^	7–10 Set × Reps ^tool^	11 Set × Reps ^tool^
Curl	3 × 15 ^ws^	3 × 15 ^ws^	2 × 15 ^d^	3 × 15 ^d^	3 × 20 ^d^
Row	2 × 15 ^ws^	2 × 15 ^ws^	2 × 10 ^d^	2 × 15 ^d^	2 × 20 ^d^
Overhead press	2 × 10 ^ws^	2 × 10 ^ws^	2 × 6 ^d^	2 × 8 ^d^	2 × 15 ^d^
Dumbbell Chest Fly	3 × 10 ^bw^	3 × 10 ^bw^	3 × 8 ^d^	3 × 10 ^d^	2 × 15 ^d^
Lateral Raise	3 × 10 ^bw^	3 × 10 ^bw^	3 × 6 ^d^	3 × 8 ^d^	2 × 20 ^d^
Triceps Kickback	2 × 10 ^bw^	2 × 10 ^bw^	2 × 6 ^d^	2 × 6 ^d^	2 × 10 ^d^
B					
Squat	2 × 10 ^bw^	2 × 10 ^bw^	2 × 10 ^d^	2 × 15 ^d^	2 × 15 ^d^ + 10″ isometric
Hip abductions	2 × 10 ^bw^	2 × 10 ^bw^	3 × 8 ^bw^	3 × 12 ^bw^	3 × 15 ^eb^
Leg Curl	2 × 10 ^bw^	2 × 10 ^bw^	3 × 8 ^bw^	3 × 12 ^bw^	3 × 12 ^eb^
Leg Extension	2 × 10 ^bw^	2 × 10 ^bw^	3 × 8 ^bw^	3 × 12 ^bw^	3 × 15 ^bw^
Sit Hip adductions	2 × 10 ^p^	2 × 10 ^p^	3 × 8 ^p^	3 × 12 ^p^	3 × 15 ^p^
Sit Calf	2 × 10 ^ws^	2 × 10 ^ws^	3 × 8 ^ws^	3 × 12 ^ws^	3 × 15 (stand)
C					
Curl	3 × 15 ^ws^	3 × 15 ^ws^	2 × 15 ^d^	3 × 15 ^d^	3 × 20 ^d^
Row	2 × 15 ^ws^	2 × 15 ^ws^	2 × 10 ^d^	2 × 15 ^d^	2 × 20 ^d^
Overhead press	2 × 10 ^ws^	2 × 10 ^ws^	2 × 6 ^d^	2 × 8 ^d^	2 × 15 ^d^
Dumbbell Chest Flys	3 × 10 ^bw^	3 × 10 ^bw^	3 × 8 ^d^	3 × 10 ^d^	2 × 15 ^d^
Lateral Raise	3 × 10 ^bw^	3 × 10 ^bw^	3 × 6 ^d^	3 × 8 ^d^	2 × 20 ^d^
Squat	2 × 10 ^bw^	2 × 10 ^bw^	2 × 10 ^d^	2 × 15 ^d^	2 × 15 ^d^ + 10″ isometric
Triceps Kickback	2 × 10 ^bw^	2 × 10 ^bw^	2 × 6 ^d^	2 × 6 ^d^	2 × 10 ^d^

Superscript letters indicate the tool used to perform the exercise: bw = body weight; d = dumbbell (two for 1 kg each one); eb = elastic band; ws = wooden stick; *p* = pillow.

**Table 2 ijerph-19-13022-t002:** Clinical and anthropometric characteristics of patients with PD (mean, SD).

Parameters	Mean ± SD
Number of participants	13
Age	64.83 ± 5.70
Height	171.58 ± 7.30
weight	76.93 ± 13.54
Disease duration	5.7 ± 3.08
Hoehn and Yahr	1.5 ± 0.5
UPDRS 1	7 ± 2.4
UPDRS 2	23.2 ± 4.5
UPDRS 3	25.2 ± 7.1
MoCA	24.5 ± 2.3
LEDD	288.5 ± 109.7

H&Y = Hoehn and Yahr; UPDRS = Unified Parkinson’s Disease Rating Scale; MoCA = Montreal cognitive assessment; LEDD = Levodopa Equivalent Daily Dose.

**Table 3 ijerph-19-13022-t003:** Motor performance and body composition variables at baseline (T0) and after 11 weeks (T1). Data from motor performance and body composition were collected at time 0 (baseline) and at the end of the 11 weeks.

	T0		T1		*p*-Value
	Mean	Std. Deviation	Mean	Std. Deviation	
Postural analysis					
Sway area OE (cm^2^)	155.63	133.96	285.57	317.60	0.30
Sway paths length OE (mm)	649.09	147.48	827.91	286.62	0.03 ^
Sway average speed OE (mm/s)	13.15	2.93	16.68	5.80	0.04 ^
Sway area CE (cm^2^)	429.90	516.67	287.15	333.63	0.30
Sway paths length CE (mm)	740.45	141.64	888.92	338.72	0.06
Sway average speed CE (mm/s)	14.90	2.70	17.87	6.84	0.06
Static analysis					
L foot area (cm^2^)	93.58	17.50	83.08	20.05	0.03 ^
R foot area (cm^2^)	94.75	21.11	88.67	23.18	0.22
R forefoot load (%)	57.75	13.47	53.75	14.16	0.10
L forefoot load (%)	47.25	12.14	40.00	12.53	0.01 *
R hindfoot load (%)	42.25	13.47	46.25	14.16	0.10
L hindfoot load (%)	52.75	12.14	60.00	12.53	0.01 *
Cervical ROM					
R lateral rotation (°)	54.98	9.14	59.84	16.47	0.18
L lateral rotation (°)	55.10	16.21	59.77	14.18	0.13
R lateral bending (°)	25.85	8.67	29.51	9.64	0.03 *
L lateral bending (°)	26.46	12.37	28.18	11.15	0.36
Flexion (°)	43.05	9.29	45.53	12.88	0.47
Extension (°)	38.62	17.70	47.77	10.98	0.49
Motor function tests					
Timed up and go (s)	12.47	2.69	9.29	2.16	0.00 ^^
5× sit-to- stand test (s)	21.86	8.52	14.19	4.22	0.00 ^^
Functional reach test (cm)	31.37	7.06	38.73	5.34	0.01 *
D peg board test (s)	137.00	49.61	126.87	51.59	0.02 ^
D hand-grip (n)	34.71	11.07	34.53	11.81	0.78
ND hand-grip (n)	31.88	10.53	33.28	11.22	0.18
D eye-hands reaction test (s)	0.54	0.36	0.41	0.11	0.07
ND foot-reaction time (s)	0.83	0.28	0.61	0.09	0.05
D foot-reaction time (s)	0.72	0.15	0.61	0.08	0.00 ^^
Body composition					
BMI	25.99	3.63	25.88	3.68	0.61
TBW (l)	39.99	6.35	40.78	8.16	0.60
FFM (Kg)	53.58	8.25	54.73	10.65	0.57
Fat (Kg)	23.33	9.34	21.84	8.61	0.45
SM (Kg)	28.57	5.80	28.62	6.61	0.95
SMI (kg/m^2^)	9.63	1.44	9.63	1.75	0.98
BMR (Kcal)	1527.33	178.18	1552.00	229.99	0.57

The significative difference between T0 and T1 for the Wilcoxon rank test it’s indicated with “^” for *p* < 0.05 and “^^” for *p* < 0.01; “*” = significative difference (*p* < 0.05) between T0 and T1 for the paired t-test; OE = open eyes, CE = closed eyes, L = left, R = right, D = dominant limb, ND = non dominant limb, BMI = body mass index, TBW = total body water, FFM = free fat mass, SM = skeletal muscle mass, SMI = Skeletal muscle mass index, BMR = basal metabolic rate.

**Table 4 ijerph-19-13022-t004:** Values of glucose metabolism, thyroid function, and bone metabolism markers at baseline and after 11 weeks of training. Blood samples were collected at time 0 (baseline) and at the end of the 11 weeks (T1).

	T0		T1		*p*-Value
	Mean	Std. Deviation	Mean	Std. Deviation	
Glucose metabolism markers					
Glycemia (mg/dL)	88.50	12.17	93.08	7.42	0.20
Insulin (mUI/L)	12.05	5.15	11.05	3.81	0.43
Insulin Sensitivity (%)	77.26	32.66	78.61	30.06	0.85
Insulin resistance	1.53	0.64	1.44	0.50	0.57
Lipid metabolism markers					
Total Cholesterol (mg/dL)	166.67	33.34	170.42	32.77	0.37
HDL Cholesterol (mg/dL)	49.33	9.59	52.08	10.13	0.11
Total HDL Cholesterol (mg/dL)	3.50	0.95	3.43	1.10	0.68
Triglycerides (mg/dL)	103.33	32.82	99.42	48.90	0.58
LDL Cholesterol (mg/dL)	96.67	30.79	98.45	30.00	0.58
LDL/HDL Cholesterol (mg/dL)	2.05	0.79	2.01	0.87	0.74
Thyroid function markers					
FT3 (ng/L)	3.25	0.70	4.64	0.47	0.00 ^^
FT4 (ng/L)	1.24	0.14	1.29	0.16	0.27
TSH (mlU/L)	1.79	1.05	1.80	0.87	0.79
Bone metabolism markers					
PTH (ng/L)	39.84	17.99	58.50	22.86	0.00 ^^
Osteocalcin (mcg/dL)	21.08	6.66	23.58	6.58	0.01 *
CTX (ng/L)	0.30	0.12	0.44	0.15	0.00 **
Vitamin D (mcg/dL)	23.38	4.83	18.58	7.04	0.04 *
Calcium (mg/dL)	8.71	0.28	9.07	0.21	0.00 **

Data are expressed as means and standard deviations. The significative difference between T0 and T1 for the Wilcoxon rank test it is indicated with “^^” for *p* < 0.01; “*” = significative difference (*p* < 0.05), “**” = significant difference (*p* < 0.01) between T0 and T1 for the paired t-test.

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
