# Peer review of "Effects of a Resistance Training Protocol on Physical Performance, Body Composition, Bone Metabolism, and Systemic Homeostasis in Patients Diagnosed with Parkinson’s Disease: A Pilot Study"

_ijerph, 2022, doi:10.3390/ijerph192013022_

Round 1

Reviewer 1 Report

Why was the right arm used for the Functional reach test and not the dominant arm?

Regarding the handgrip test, use a neutral term rather than “he” for the subject.

How was the strength training protocol determined; exercises, volume, intensity, frequency? How was the increase in intensity determined? There is no reference to any prior effectiveness of the strength training protocol used. Was the literature reviewed examining strength training protocols for people with Parkinson’s disease?

Author Response

Dear reviewer, attached you will find the file with the answers to your questions. thank you for your comments, they helped to improve the quality of our work.

Reviewer 2 Report

Thank you for submitting your manuscript to the International Journal of Environmental Research and Public Health, your manuscript requires a number of important and substantial changes.

The article requires an in-depth review of the English language by an expert.

Indicate the type of study in the title, being well defined, it is not clear if it is a protocol or a quasi-experimental study.

The title is confusing, simplify the content and clearly define the purpose of the study. As a suggestion: Effects of a resistance training protocol on physical performance, body composition, bone metabolism and systemic homeostasis in patients diagnosed with Parkinson's disease: A quasi-experimental study.

Reduce and simplify the number of Keywords according to the MeSH terms, I think they are not the most appropriate for the subject of the manuscript, for example Parkinson's disease should be a term that should appear since it is about the study population that is the subject of the manuscript.

Reformulate and rewrite the abstract with the following sections, they should be written in the form of: Background. This is a concise statement of the reasons for conducting this research, placing it in the context of current knowledge or controversies. Objective. This is a clear statement of the precise objective or question being addressed in the paper this may take the form of objectives or contrasting hypotheses. Methods. The basic design of the study and its duration should be described for quantitative and qualitative research quantitative and qualitative the methods used should be indicated and the data/statistical methods should be provided. Results. The main results of the study should be stated in narrative form any measurements or other information that may require explanation information that may require explanation confidence intervals are preferred to p-values values confounders, modifiers or mediators, as well as any other factors crucial to the outcome of the study should be indicated crucial to the outcome of the study should be indicated. Conclusions. The conclusions of the study that are directly supported by the evidence presented should of the evidence reported, along with clinical application, and speculation about the potential impact on current thinking current thinking.

In the introduction, they should give a detailed and comprehensive description of the study population; incidence; classification; diagnosis; current treatments; physical, psychological, social, and economic problems associated with the injury, as recommended by the Parkinson's Disease Guidelines (an important aspect of the rationale and importance of the study) in both the introduction and discussion.

Report a checklist according to the type of study they present.

Indicate in the methodology the study design used to execute this project / Context.  Describe the relevant setting, locations and dates, including periods of recruitment, exposure, follow-up and data collection.

Clearly define the exclusion criteria of the participants which are left to chance, they cannot indicate the following "participants were excluded if they were present at less than the 80% of the scheduled training sessions" this is a manipulation of the sample, they have to clearly define the criteria for inclusion and exclusion of participants and take into consideration how they are going to assume the loss of subjects from the study. Define how you are going to assume the losses. If this was done in this way it increases the risk of bias of your study you should indicate it in the discussion.

10º The sample size is very small. Why is the number of participants so small? How did you solve this problem? How was the sample size determined?

11º You could provide a flow chart or images of the exercises performed by the participants.

12º Specify all measures taken to address potential sources of bias.

13º Must justify with literature the Motor performance tests Postural and static analysis and Cervical ROM.

14º Table 1 of the characteristics of the participants should be included in the results section and all the data obtained, age, weight, height, etc. should be reflected in addition to the data provided and the number of participants.

15º In Table 3 and 4, T0 (baseline) and T1 (11 weeks) indicate the measurement period, because long-term follow-up was not performed since Parkinson's disease is chronic in nature, which increases the risk of bias in the study, as well as the small number of participants or the lack of a control group, which has not been included in the discussion within the limitations of the study.

16º The limitations of the study, which are several, should be specified in the discussion.

17º Discussions should cover the key findings of the study: discuss any previous research related to the topic to place the novelty of the discovery in the appropriate context, discuss possible shortcomings and limitations in its interpretations, discuss its integration into the current understanding of the problem and how. This advances current views, speculates on the future direction of research, and freely postulates theories that could be tested in the future, completed, and reformulated. The discussion should be rewritten to present serious errors.

18º Prepare a specific section for conclusions.

19º The bibliography is not adapted to the standards of the journal. The citations of the references in the text are erroneous, an in-depth revision of this aspect is needed, I recommend the authors to review the author's guide of the journal. Use the reference style, based on the American Chemical Society (ACS) style, both in the text and in the bibliography section, as indicated in the author's guide of the journal.

Author Response

(The authors gave the same response as above.)

Reviewer 3 Report

Hello,

I enjoyed reading your paper and feel it is very worthwhile work. However, I have some significant questions you may wish to consider.

I wish you all the best.

* Page 1- "The alterations of resorption-formation mechanism balance are frequently associated with other pathologies, like neurodegenerative disorders, characterized by motor impairment such as walking instability, rigidity, tremors, bradykinesia [2] as in Parkinson’s disease (PD), in which it’s demonstrated that motor symptoms are associated with a decrease in bone mineral density [3]."- PLEASE CONSIDER REVISING TO MAKE MORE CONCISE

* Page 1- "Physical activity is a key stimulus for symptoms management but it also produces lactate and myokines that contribute to brain health

* Page 2- "Exercise seems to be effective if it has certain characteristics such as impact with the ground or other surfaces [17] EFFECTIVE FOR WHAT?

*Page 2- "In addition, a mechanism that explains the importance of mechanical loading is that this loading, through mRNA stimulation, regulates the activity of osteogenic and bone resorption factors while maintaining the properties of the tissue

* Page 2- " The loss of mechanical stimulus is the main cause of bone disease in PD due to decreased daily life physical activity. Typically, people with PD, people with PD frequently adopt a sedentary lifestyle as a result of their reduced postural stability, gait disturbances, and decreased strength."

Page 2- "Moreover, it is proved has been proven, has been shown, that one of the possible causes of the onset of PD is genetic [24-26]

Page 2-" In fact, some genes whose altered expression is linked to PD have been identified (the PARKS) PLEASE SPELL OUT ACRONYM FIRST TIME

Page 2- "United Kingdom Parkinson society Brain Bank" PLEASE REFERENCE

Page 3- PLEASE REFERENCE ALL STANDARDIZED TESTING AT TOP OF PAGE 3

- Page 6- did each participant use the same strength elastic band? Did the participants continue with 1 kg dumbells for the length of the intervention?

- Curious as to why you chose to measure cervical ROM and not something like lower extremity flexibility or strength, given that a vast majority of your interventions do not directly influence the neck?

Author Response

(The authors gave the same response as above.)

Round 2

Reviewer 2 Report

Congratulations to the authors for the corrections.

1º Make formatting corrections/modifications. 

2º Clearly define in the title of the manuscript whether they conducted a pilot study or a feasibility study, not both. 

3º The language needs to be revised.

Author Response

Dear reviewer ,

thank you for your comments, we really believe that they have contributed to the improvement of our work. In the latest version of the manuscript, we have removed the word "feasibility" and kept only "pilot" also we have again had the manuscript proofread by a mother tongue English speaker (corrections are in red in the text). Thank you again.

Best regards